# A Study on the Measurement and Influences of Energy Green Efficiency: Based on Panel Data from 30 Provinces in China

**Yulin Lu [1,†], Chengyu Li [2,†] and Min-Jae Lee [3,\*]**

1   Directly Affiliated College, Shandong Open University, Jinan 250014, China; luyulin0622@outlook.com
2   College of Economics and Management, Shandong University of Science and Technology,
    Qingdao 266590, China; woailaoer5555@outlook.com
3   Department of Global Business, Mokwon University, Daejeon 35349, Republic of Korea
\*   Correspondence: mjlee@mokwon.ac.kr
†   These authors contributed equally to this work.

**Abstract:** China's rapid economic growth has inevitably led to serious resource depletion, environmental degradation, and a decline in social welfare. As such, establishing total-factor energy green efficiency (TFEGE) and exploring its factors are of paramount importance to bolster comprehensive energy efficiency and foster sustainable development. In this research, we deployed the spatial lag model (SLM) and data envelopment analysis (DEA), using energy, capital and labor as input indicators, GDP and social dimension metrics as desirable outputs, and "three wastes" as undesirable outputs, to assess the TFEGE across 30 provinces in China from 2001 to 2020. Employing the exploratory spatial data analysis (ESDA) method, we analyzed the spatial autocorrelation of TFEGE at national and provincial levels. Simultaneously, we examined the influencing factors of TFEGE using a spatial econometric model. Our study reveals that, throughout the examined period, the TFEGE in China has generally shown a steady decline. The TFEGE dropped from 0.630 to 0.553. The TFEGE of all regions in China also showed a downward trend, but the rate of decrease varied significantly across different regions. Among them, the TFEGE of the eastern region fluctuated between 0.820 and 0.778. The TFEGE of the northeast region decreased significantly from 0.791 to 0.307. The TFEGE of the western region decreased from 0.512 to 0.486. The TFEGE of the central region decreased from 0.451 to 0.424. Beijing, Guangdong, Hainan, Qinghai, and Ningxia showed an effective TFEGE, while for other provinces, it was ineffective. The TFEGE in all four major regions failed to achieve effectiveness. Its distribution pattern was east > west > northeast > central. The TFEGE across the 30 provinces showed positive spatial autocorrelation, indicating a strong spatial clustering trend. We found that while transportation infrastructure and technological progression exert a positive impact on TFEGE, elements such as industrial structure, energy composition, and foreign direct investment negatively influence TFEGE.

**Keywords:** total-factor energy green efficiency; sustainable development; exploratory spatial data analysis; data envelopment analysis

## 1. Introduction

In late 2008, the United Nations Environment Programme [1] introduced the concept of a "green economy," emphasizing its important role in economic growth. It urged nations worldwide to develop a green economy, aiming for a transforming economic growth mode to better address the various challenges of sustainable development. After the reform and opening-up policies were adopted in 1978, China shifted its economy from the government-directed, single-planned model to a dual system of planned and market economies [2]. This transformation led to China's rapid economic growth, which led it to be ranked as the second-largest economy worldwide by 2022 [3]. However, this economic expansion has come at the expense of substantial resource and energy consumption. Jiang

and Shi [4] constructed the Enhanced Sustainable Development Index (ESDI) to measure the level of regional sustainable development in China and found that China's overall ESDI has declined, with large regional differences, and the spatial inequality in development sustainability continues to intensify. This indicates that ecological imbalances and the deterioration of living conditions are continuing due to the prolonged development of high pollution, high consumption and low efficiency. Changing the economic growth strategy, reducing energy consumption and improving energy efficiency are necessary for China to overcome hurdles that constrain the sustainability of the economy, society and the environment, and to realize economic transformation and high-quality development.

Energy is a source of power. Its output is dependent on production factors, such as energy itself. Energy efficiency refers to the conversion rate of energy, emphasizing the optimization of benefits from minimal energy input [5]. Thus, energy efficiency denotes the ratio of inputs firm production factors, such as energy, to their output during production [6]. As we have seen above, enhancing energy efficiency in the current Chinese economy is pivotal for China's sustainable growth and is a key consideration in policymaking for industrial structural transformation. It is also an important strategy for promoting green technological innovation and achieving multiple goals, such as energy conservation and emission reduction, China's ecological civilization construction and green sustainable development [7]. Accordingly, as energy green efficiency reflects sustainability and acts as an important indicator for technological advancement, industrial advancement and social development, research aiming to measure and evaluate its impact is becoming increasingly important. Consequently, to increase energy efficiency and drive green growth, it is imperative to develop a comprehensive and precise evaluation index for energy efficiency.

For evaluation indicator development, this study defines energy green efficiency as the input of energy and other related factors of production to economic, environmental, and social output based on energy efficiency and the green development concept [8]. The concept of energy efficiency encompasses three dimensions: economic, environmental, and social. The economic dimension emphasizes maximizing economic output with minimal energy and economic input. The environmental dimension requires awareness of the detrimental impacts of consuming energy and resources on the ecosystem, striving to gradually diminish the damage to the environment caused by harmful outputs like $CO_2$. The social dimension underscores a people-first approach to energy and resource utilization, ensuring that energy consumption fulfills the people's material and spiritual needs, embodies shared and equitable distribution, elevates the level of social welfare, and enhances overall well-being [9].

Meanwhile, existing studies have had the following issues in their empirical analyses. First, energy efficiency is a desirable outcome where the inputs of capital, labor, and energy consumption remain unchanged. However, previous studies measuring this aimed to maximize economic output by focusing on gross domestic product (GDP) [10,11], so consideration of social and environmental factors was largely ignored. To the best of our knowledge, there have been no studies that have constructed sustainable energy green efficiency indicators by incorporating economic, environmental, and social indicators. Therefore, in deriving energy green efficiency indicators, we include minimizing environmental pollutants and optimizing social services while considering economic output.

Second, research on energy efficiency has been ongoing for many years worldwide and has become a hot topic in the field of energy economics. The study of energy efficiency is primarily based on single-factor or multifactor research frameworks. Under the single-factor research framework, energy efficiency is mainly represented by energy productivity (i.e., the ratio of GDP to a certain related input factor) or energy intensity (i.e., the ratio of energy input to output). Because the measurement of energy efficiency is relatively simple under a single-factor framework, some scholars, such as [12,13], have used this approach to study energy efficiency. However, this method has its drawbacks. It tends to overestimate energy efficiency and overlook the substitutability among various variables, failing to meet research demands. Therefore, Huang and Wang [14] proposed an energy

efficiency estimation method based on the multifactor framework, which considers not only energy input variables but also other related ones. For example, Chang and Hu [15] also adopted the multifactor method for their energy efficiency research, based on Huang and Wang's study. Moreover, with sustainable issues emerging as the center of China's development, studies that have taken overall measurements based on valid indicators are still in their infancy.

This study can contribute twofold by filling in the research gaps. First, this study integrates the UN-SLM-DEA model to develop indicators that can identify energy green efficiency output in terms of sustainability. The developed framework can extend from the economic sector to the environmental sector, and further to the social sector to contribute to measuring energy green efficiency. Second, based on the developed indicators, we will measure and analyze TFEGE in 30 provinces in China to reveal the true level and regional differences in TFEGE in those provinces. By constructing a spatial econometric model to empirically study the spatial effects and influencing factors of TFEGE, we can identify the driving force for improving TFEGE. On this basis, targeted policy suggestions are proposed to provide a reference for the government and relevant departments to formulate policy measures.

## 2. Literature Review

The most widely used multifactor energy efficiency measurement method is DEA. As a non-parametric analysis method, DEA is accurate, comparable, easy to decompose, and effectively reflects efficiency differences between different decision units [16]. Since the introduction of the DEA method, many scholars worldwide have developed various DEA models to measure efficiency, particularly in the field of energy efficiency. For instance, Wang et al. [17] studied the energy efficiency of different regions in China using the Meta-frontier DEA method. Wang et al. [18] evaluated China's industrial total factor energy efficiency (TFEE) using a two-stage DEA model with shared inputs. Zhao et al. [19] assessed the TFEE of BRICS countries based on a three-stage DEA model. Xiao et al. [20] used the S-U-SLM model to measure sectoral energy environmental efficiency. Mocholi-Arce et al. [21] used the Double-Bootstrap method to evaluate the energy efficiency of drinking water treatment plants. Chen et al. [22] employed the Windows-DEA model to calculate and analyze the transport energy efficiency in cities of China's Yangtze River Delta. Yang and Wei [23], based on the Game Cross-Efficiency DEA, calculated and analyzed the TFEE of cities in China from the perspective of environmental pollution. Uddin et al. [24] conducted a study on the sustainability of green taxation and energy efficiency in Bangladesh. Wang et al. [25] investigated the impact of China's low-carbon urban policies on energy efficiency. Shehzadi [26] conducted research on energy efficiency and productivity in emerging and developing countries in Asia. Moreover, Khazaei et al. [27] conducted a systematic study on the types of renewable energy used in power generation in Iran, and Khazaee et al. [28] evaluated the capacity of Asian countries to produce renewable energy.

At the same time, energy efficiency research is no longer limited to simple efficiency measurements and has shifted towards studying more complex issues. For example, Yu et al. [29] calculated the industrial energy efficiency of Chinese cities and used the Local Moran's Index to analyze the spatial clustering characteristics of industrial energy efficiency. Pan et al. [30] used the DEA model to calculate the regional TFEE, and applied the Markov Chain and Spatial Markov Chain to test the convergence of energy efficiency in China. Lv et al. [31], from the perspective of spatial effects, used the Windows-DEA model to evaluate China's regional dynamic energy efficiency from 2001 to 2010, and explored the determinants of energy efficiency through a spatial econometric model. He et al. [32] combined the DEA model, rough set theory and fuzzy artificial neural networks to explore the factors influencing energy efficiency. Yang and Wei [33] analyzed the TFEE of 17 core regions of countries along the "Belt and Road" under environmental constraints from 2005 to 2015 and used the Malmquist Index and Tobit model to examine the internal and external

factors influencing TFEE. In summary, scholars have further explored energy efficiency based on aspects like spatial characteristics, convergence, and influencing factors.

The scope of energy efficiency research is very broad, including national, regional, city, industry, and sector dimensions. For instance, Zhang et al. [34] and Zhao et al. [19] respectively studied the TFEE of 23 developing countries and 35 Belt and Road countries. Honma and Hu [35] calculated the regional TFEE of Japan from 1993 to 2003. Honma and Hu [36] also measured and analyzed the energy efficiency of Japan's 17 economic sectors from 1998 to 2015 based on the total factor framework. Huang and Wang [14] evaluated and analyzed the TFEE of 276 Chinese cities from 2000 to 2012 from a dual perspective of management and environment. Liu et al. [37] evaluated and analyzed the TFEE of China's thermal power industry from both static and dynamic perspectives.

From the above analysis, it is clear that research on energy efficiency is becoming more mature. However, there are still areas for improvement. (1) There is a lack of consideration for social welfare. Current research on energy efficiency is confined to the efficiency between maximizing economic output and minimizing environmental pollution, without considering social welfare in the research framework. This means extending energy efficiency to energy green efficiency that ensures the maximization of economic output, the minimization of environmental pollution, and the optimization of social welfare. (2) There is a lack of research on the spatial autocorrelation of energy efficiency. (3) Most studies on the factors influencing energy efficiency are based on traditional econometric models, with relatively few studies using spatial econometric models to investigate these factors.

Based on the above analysis, this study uses panel data from 30 provinces in China as a foundation and employs the UN-SLM-DEA model to evaluate the total factor energy green efficiency (TFEGE) of these provinces from 2001 to 2020. The ESDA method is also used to analyze the spatial autocorrelation of TFEGE at different levels. It further employs the panel spatial autoregressive (SAR) model to study the factors influencing TFEGE. Finally, empirical results are obtained with policy suggestions.

## 3. Methods

### 3.1. The SLM-DEA Model

DEA is a nonparametric analysis method that uses linear programming ideas to evaluate the relative effectiveness of comparable types of decision-making units. This technique has been widely used in many fields and has become one of the most popular technical tools for evaluating relative efficiency [38]. The DEA measurement model is mainly divided into four categories [39]: ① radial, angle; ② radial, non-angle; ③ non-radial, angle; ④ non-radial, non-angle. Radial means that the input or output changes in the same proportion when measuring efficiency to achieve efficiency. The angle indicates the input or output angle when measuring efficiency. The traditional DEA models such as CCR and BCC mostly measure efficiency based on the radial and angle measurements. However, they fail to fully consider the slack problem of input and output and cannot accurately measure the efficiency value considering undesirable output. In response to this situation, Tone proposed a non-radial, non-angle SLM model that incorporates slack variables directly into the objective function. In spatial econometrics, one sometimes sees all-encompassing specifications involving various autoregressive spatial lags [40]. This not only solves the problem of slack in input and output, but also greatly improves the accuracy of the measurement results after considering the undesirable outputs [41]. In this vein, this study utilizes the DEA-SLM model, which is a measure of non-parameter efficiency considering non-expected outputs. Through the relationship between each input and output and non-expected output, we study the efficiency between the decision unit and the front efficiency plane. In this study, we used the SLM model to study the total factor green energy efficiency of 30 provinces, and considering both positive and negative outputs, we made the results more scientific and rational. Therefore, this paper constructs an UN-SLM model that considers undesirable outputs to measure China's green energy efficiency more accurately, as shown in Equation (1)

$$\rho^* = \min \frac{1 - \frac{1}{I}\sum\limits_{i=1}^{I}\frac{s_i^x}{x_{k'i}^{t'}}}{1 + \frac{1}{M+N}\left(\sum\limits_{m=1}^{M}\frac{s_m^y}{y_{k'm}^{t'}} + \sum\limits_{n=1}^{N}\frac{s_n^z}{z_{k'n}^{t'}}\right)}$$

$$s.t. \begin{array}{l} \sum\limits_{t=1}^{T}\sum\limits_{k=1}^{K}u_k^t x_{ki}^t + s_i^x = x_{k'i}^{t'} \quad (i = 1, \ldots, I) \\[2mm] \sum\limits_{t=1}^{T}\sum\limits_{k=1}^{K}u_k^t y_{km}^t - s_m^y = y_{k'm}^{t'} \quad (m = 1, \ldots, M) \\[2mm] \sum\limits_{t=1}^{T}\sum\limits_{k=1}^{K}u_k^t z_{kn}^t + s_n^z = y_{k'n}^{t'} \quad (n = 1, \ldots, N) \\[2mm] u_k^t \geq 0, s_i^x \geq 0, s_m^y \geq 0, s_n^z \geq 0 \; (k = 1, \ldots, K) \end{array} \tag{1}$$

where $\rho^*$ is the objective function value; $u_k^t$ is the weight vector; $I$, $M$ and $N$ are the number of inputs, desirable outputs and undesirable outputs, respectively; $s_i^x$, $s_m^y$ and $s_n^z$ are the relaxation vectors of inputs, desirable outputs and undesirable outputs, respectively; $x_{i'}^{t'}$, $y_{m'}^{t'}$ and $z_{n'}^{t'}$ are inputs, desirable outputs and undesirable outputs during the $k'$th production of unit $t'$. $\rho^*$ is strictly monotonically decreasing about $s_i^x$, $s_m^y$ and $s_n^z$, and have $0 < \rho^* \leq 1$. If $\rho^* = 1$, it means that $s_i^x$, $s_m^y$ and $s_n^z$ are all 0, and the decision unit is valid. If $\rho^* < 1$, it means that the decision-making unit is invalid. Inputs and outputs need to make the necessary improvements.

### 3.2. Spatial Analysis Methods

Typically, the analysis process in spatial econometrics consists of two main stages. In the initial stage, a spatial autocorrelation test is conducted to investigate whether a spatial correlation exists amongst the dependent variables. This step is vital because the application of spatial econometric analysis is only warranted when a spatial autocorrelation is present. For this part of the analysis, we utilize GeoDa 1.6 software (Informer Technologies, Inc., Los Angeles, CA, USA; please refer to Section 3.2.1 for details). Subsequently, the presence of spatial autocorrelation informs the selection of an appropriate spatial econometric model. This model is then used to analyze the relationships between dependent and explanatory variables. MATLAB software version R2023b (MathWorks, Natick, MA, USA) is utilized for this stage of the analysis (for a more detailed explanation, please see Section 3.2.2).

#### 3.2.1. Spatial Autocorrelation Test

Before conducting spatial econometric analysis, this paper conducts spatial autocorrelation test on TFEGE of 30 provinces. To achieve this, the global Moran's *I* index is used, as shown in Equation (2)

$$Moran's\ I = \frac{\sum\limits_{i=1}^{n}\sum\limits_{j=1}^{n}W_{ij}(Z_i - \overline{Z})(Z_j - \overline{Z})}{S^2\sum\limits_{i=1}^{n}\sum\limits_{j=1}^{n}W_{ij}} \tag{2}$$

where $n$ is the number of research objectives; $Z_i$ and $Z_j$ are observations of the $i$ and $j$ regions; $W_{ij}$ is the spatial weight matrix (space adjacent to 1, not adjacent to 0); $S^2$ is the observed variance; $\overline{Z}$ is the average of the observations. The Moran's *I* index range is $[-1,1]$, the closer to 1 or $-1$, the stronger the spatial correlation. If the Moran's *I* index is positive, it indicates that agglomeration is present. If the Moran's *I* index is negative, it indicates that the representation is spatially different. When the Moran's *I* index is close to 0, the spatial distribution is random and there is no spatial correlation.

### 3.2.2. Spatial Econometric Models

The spatial econometric model can effectively solve the spatial dependence and spatial correlation between the variables being investigated. It is mainly divided into two types: SLM and spatial error model (SEM) [42].

The SLM can be specified as:

$$y = \rho Wy + X\beta + \varepsilon \tag{3}$$

where $y$ is the dependent variable; $X$ is the explanatory variable; $W$ is the spatial weight matrix; $\varepsilon$ is a random error term; $\rho$ is the spatial regression coefficient; $\beta$ is an estimated independent coefficient.

The SEM can be written as:

$$\begin{aligned} y &= X\beta + \varepsilon \\ \varepsilon &= \lambda W\varepsilon + \nu \end{aligned} \tag{4}$$

where $y$ is the dependent variable; $X$ is the explanatory variable; $W$ is the spatial weight matrix; $\varepsilon$ is a random error term; $\lambda$ is the spatial autocorrelation coefficient; $\beta$ is an estimated independent coefficient; $\nu$ is a disturbance term.

Thus, the specific forms of SLM and SEM utilized in this study are as follows:
The SLM:

$$\begin{aligned} TFEGE_{i,t} &= \beta_0 + \rho\sum_{j=1}^{30} W_{ij}TFEGEI_{i,t} + \beta_1 InINT_{i,t} + \beta_2 InIS_{i,t} + \\ &\quad \beta_3 InES_{i,t} + \beta_4 InTP_{i,t} + \beta_5 InFDI_{i,t} + \beta_6 InEA_{i,t} + \beta_7 InGZ_{i,t} + \varepsilon_{i,tz} \\ &\varepsilon_{i,t} \sim N(\sigma^2_{i,t}) \end{aligned} \tag{5}$$

The SEM:

$$\begin{aligned} TGEGE_{i,t} &= \beta_0 + \beta_1 InINT_{i,t} + \beta_2 InIS_{i,t} + \beta_3 InES_{i,t} \\ &\quad +\beta_4 InTP_{i,t} + \beta_5 InFDI_{i,t} + \beta_6 InEA_{i,t} + \beta_7 InGZ_{i,t} + \varepsilon_{i,t} \\ &\varepsilon_{i,t} = \lambda\sum_{j=1}^{30} W_{ij}\varepsilon_{i,t} + v_{i,t}, v_{i,t} \sim N(\sigma^2_{i,t}) \end{aligned} \tag{6}$$

In Formulas (5) and (6), the variables are defined as natural logarithms; $i$ is the province; $t$ is the year; $\beta_0$ is the intercept term: the parameters $\beta_i$, $i = 1, 2, 3 \ldots 7$ are the undetermined coefficients of all variables, respectively; $W_{ij}$ is the spatial weight matrix; $\rho$ is the spatial regression coefficient which indicates the impact of the TFEFE of the nearby regions on this region in this study; $\lambda$ is the spatial autocorrelation coefficient which indicates the impact of the residuals of nearby regions on the residuals of this region; $\varepsilon_{i,t}$ is the error term; $v_{i,t}$ is the random error term, which obeying the normal distribution.

## 4. Data and Variable Description

This study utilizes the UN-SLM model to evaluate the Total-factor Energy Green Efficiency (TFEGE) of 30 provinces in China (including 4 municipalities, 4 autonomous regions, and 22 provinces) from 2001 to 2020. Due to data availability, Tibet, Hong Kong, Macao, and Taiwan are not included in the study. For regional analysis convenience, China was divided into four economic regions based on the approaches of Chengyu et al. [43], as shown in Figure 1. According to the definition of TFEGE, we consider energy, capital and labor as inputs; Gross Domestic Product (GDP) and social dimension indicators (natural population growth rate, urban population ratio, proportion of educational expenditure in fiscal expenditure, average years of schooling, proportion of health expenditure in fiscal expenditure, doctors per thousand people, hospital beds per thousand people, and average life expectancy) as desirable outputs; and three wastes (waste gas, wastewater, and solid wastes) as undesirable outputs. The Exploratory Spatial Data Analysis (ESDA) method is used to analyze the spatial autocorrelation of TFEGE at different levels. In addition, the Spatial Autoregressive (SAR) model is adopted to investigate the factors influencing TFEGE. The study concludes with empirical results and policy recommendations. The data primarily come from the "China Statistical Yearbook (2002–2021)", "China Energy Statistical

Yearbook (2002–2021)", "China Environmental Yearbook (2002–2021)", and "China Health Statistics Yearbook (2002–2021)", supplemented with statistical data from the 30 provinces.

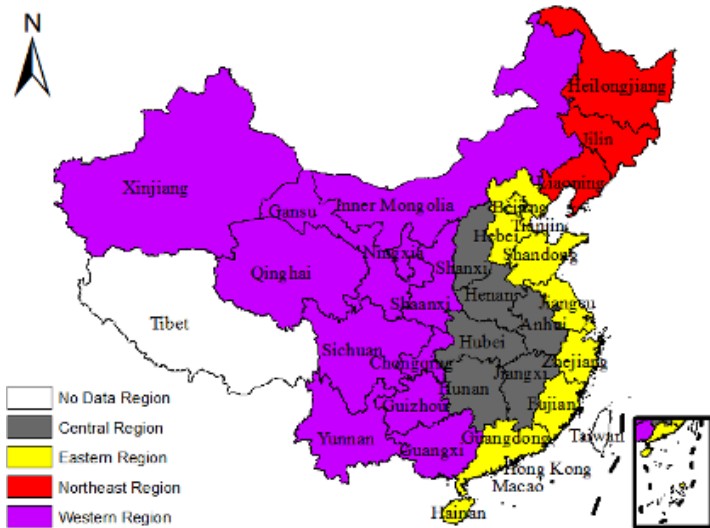

**Figure 1.** Four economic regions of China.

In this study, the evaluation indicators of TFEGE include three parts: input, desirable output and undesirable output indicators. The factors influencing TFEGE include seven variables: transportation infrastructure (TI), industrial structure (IS), energy structure (ES), technological progress (TP), foreign direct investment (FDI), economic agglomeration (EA), and environmental regulation (GZ). All variables used in this paper are shown in Table 1, with specific explanations as follows.

**Table 1.** Variable description.

| Variable | Index | Unit |
|---|---|---|
| TFEGE | Total-factor Energy Green Efficiency | |
| Energy | Energy Consumption | 10,000 tons of SCE |
| Capital | Capital Stock Based 2001 Year | 100 million yuan |
| Labor | Total Employment Population | 10,000 persons |
| GDP | Gross National Product Based 2001 Year | 100 million yuan |
| Population Control | Natural Population Growth Rate | % |
| Urbanization Level | Urban Population Ratio | % |
| Science and technology Attention Degree | Science & Education expenditure/Fiscal Expenditure | % |
| High Quality Population | Average Years of Schooling | year |
| Medical Attention Degree | Health Expenditure/Fiscal Expenditure | % |
| Medical Resources Level | Number of Doctors per Thousand People | persons |
| | Number of Beds per Thousand People | |
| Health Level | Life Expectancy per Capita | year |
| Waste Gas | Carbon Dioxide Emissions | 10,000 tons |
| | Sulfur Dioxide Emissions | 10,000 tons |
| | Soot and Dust Emission | 10,000 tons |
| Wastewater | Wastewater Discharged | 10,000 tons |
| | Chemical oxygen Discharged | 10,000 tons |
| | Ammonia Nitrogen Discharged | 10,000 tons |
| Solid Wastes | Industrial Solid Wastes Discharged | 10,000 tons |
| Transportation infrastructure | Railway Density | km/km$^2$ |
| Industrial Structure | Secondary Industry Output Value/GDP | % |

**Table 1.** *Cont.*

| Variable | Index | Unit |
|---|---|---|
| Energy Structure | Coal Consumption/Energy Consumption | % |
| Technological Progress | The Number of Patents Granted | 1Pcs |
| Foreign Direct Investment | Foreign Direct Investment/GDP | % |
| Economic Agglomeration | Non-Agricultural Output Per Unit Area | 100 million yuan/km$^2$ |
| Environmental Regulation | Environmental Pollution Control Investment/GDP | % |

(1) Input indicators include energy consumption, labor stock, and capital stock. The data on energy consumption are directly taken from the China Energy Statistics Yearbook. The capital stock is calculated based on the Perpetual Stock Method of 2000. The calculation formula of capital stock is: $K_t = I_t + (1 - \delta_t)K_{t-1}$. The formula for calculating the labor stock is: (number of employees at the end of the year + number of employees at the end of the previous year)/2.

(2) Desirable output indicators include economic indicators such as GDP and social-related indicators. GDP is calculated based on the statistics from 2000. Social dimension indicators include population control rate, urbanization level, science and technology attention degree, high-quality population, medical attention degree, medical resource level, and health level [29]. The population control rate is represented by the natural population growth rate. The urbanization level is represented by the urban population ratio. The science and technology attention degree is represented by the ratio of science & education expenditure to fiscal expenditure. The high-quality population is represented by the average years of schooling. The medical attention degree is represented by the ratio of health expenditure to fiscal expenditure. The medical resource level is represented by the number of beds and doctors per thousand people. The health level is represented by life expectancy per capita.

(3) Undesirable output indicators include waste gas, wastewater, and solid wastes. Waste gas includes emissions of carbon dioxide, sulfur dioxide, and industrial soot and dust. Wastewater includes wastewater discharge, chemical oxygen demand (COD) emissions and ammonia nitrogen emissions. Solid wastes include the volume of industrial solid wastes discharged. Carbon dioxide emissions are estimated by the following formula: energy consumption × standard coal conversion coefficient × carbon emission coefficient. Other undesirable output data comes from the China Environmental Statistics Yearbook.

(4) Transportation infrastructure is represented by railway density. Industrial structure is represented by the ratio of secondary industry output value to GDP. Energy structure is represented by the ratio of coal consumption to energy consumption. Technological progress is represented by the number of patents granted. Foreign direct investment is represented by the ratio of foreign direct investment to GDP. Economic agglomeration is represented by the non-agricultural output per unit area. Environmental regulation is represented by the ratio of investment in environmental pollution control to GDP.

At the same time, DEA measurement requires two conditions to be met: ① The number of data samples must be more than three times the sum of input and output indicators. ② The number of data samples must be greater than the product of input and output indicators. As the annual sample number of this study is limited, and the number of social and environmental dimension indicators is too large, it is impossible to meet the requirements of DEA measurement. Therefore, this study refers to the processing method of [44], using the Principal Component Analysis (PCA) method for dimension reduction. The indicators of the social dimension and environmental dimension are each merged into a single indicator to ensure that the number of data samples meets the DEA measurement conditions.

## 5. Results and Discussion

### 5.1. Estimate of the Total-Factor Energy Green Efficiency

Based on the SLM model that considers undesirable outputs, we have calculated the Total-factor Energy Green Efficiency (TFEGE) for each province and region from 2001 to 2020. The results are shown in Figure 2, Tables 2 and 3.

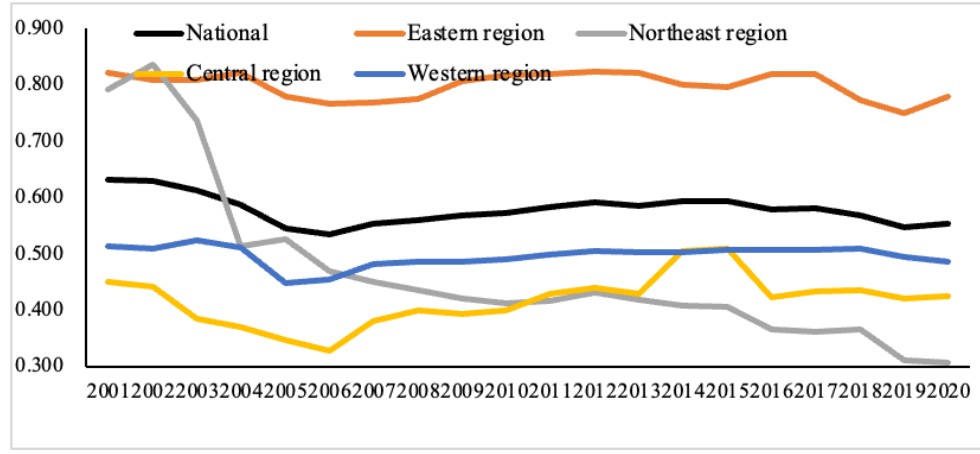

**Figure 2.** China and regions total-factor energy green efficiency changing trend (2001–2020).

**Table 2.** Provincial total-factor energy green efficiency.

| Province | 2001 | 2010 | 2020 | Mean | Ranking |
|---|---|---|---|---|---|
| Beijing | 1.000 | 1.000 | 1.000 | 1.000 | 1 |
| Guangdong | 1.000 | 1.000 | 1.000 | 1.000 | 2 |
| Hainan | 1.000 | 1.000 | 1.000 | 1.000 | 3 |
| Qinghai | 1.000 | 1.000 | 1.000 | 1.000 | 4 |
| Ningxia | 1.000 | 1.000 | 1.000 | 1.000 | 5 |
| Tianjin | 1.000 | 1.000 | 1.000 | 0.964 | 6 |
| Shanghai | 1.000 | 1.000 | 0.748 | 0.955 | 7 |
| Jiangsu | 0.675 | 1.000 | 1.000 | 0.863 | 8 |
| Zhejiang | 0.727 | 0.742 | 0.735 | 0.728 | 9 |
| Fujian | 1.000 | 0.491 | 0.581 | 0.612 | 10 |
| Jiangxi | 1.000 | 0.456 | 0.442 | 0.575 | 11 |
| Shandong | 0.498 | 0.560 | 0.484 | 0.536 | 12 |
| Jilin | 0.850 | 0.407 | 0.345 | 0.508 | 13 |
| Liaoning | 1.000 | 0.425 | 0.278 | 0.473 | 14 |
| Chongqing | 0.372 | 0.477 | 0.553 | 0.466 | 15 |
| Inner Mongolia | 1.000 | 0.390 | 0.295 | 0.459 | 16 |
| Gansu | 0.355 | 0.376 | 0.402 | 0.426 | 17 |
| Heilongjiang | 0.522 | 0.404 | 0.299 | 0.426 | 18 |
| Guangxi | 0.553 | 0.372 | 0.323 | 0.408 | 19 |
| Xingjiang | 0.527 | 0.458 | 0.304 | 0.408 | 20 |
| Hunan | 0.276 | 0.430 | 0.454 | 0.403 | 21 |
| Shanxi | 0.484 | 0.366 | 0.319 | 0.395 | 22 |
| Hubei | 0.280 | 0.382 | 0.468 | 0.388 | 23 |
| Anhui | 0.301 | 0.401 | 0.436 | 0.384 | 24 |
| Shannxi | 0.292 | 0.406 | 0.359 | 0.377 | 25 |
| Henan | 0.365 | 0.363 | 0.426 | 0.356 | 26 |
| Hebei | 0.295 | 0.369 | 0.236 | 0.323 | 27 |
| Sichuan | 0.150 | 0.290 | 0.452 | 0.312 | 28 |
| Guizhou | 0.138 | 0.322 | 0.367 | 0.305 | 29 |
| Yunnan | 0.249 | 0.296 | 0.295 | 0.293 | 30 |
| Mean | 0.630 | 0.573 | 0.553 | 0.578 | |

**Table 3.** Regional total-factor energy green efficiency.

| Region | 2001 | 2006 | 2011 | 2016 | 2020 | Mean | Ranking |
|---|---|---|---|---|---|---|---|
| Eastern region | 0.820 | 0.765 | 0.819 | 0.818 | 0.778 | 0.798 | 1 |
| Western region | 0.512 | 0.454 | 0.498 | 0.507 | 0.486 | 0.496 | 2 |
| Northeast region | 0.791 | 0.469 | 0.416 | 0.365 | 0.307 | 0.469 | 3 |
| Central region | 0.451 | 0.327 | 0.429 | 0.421 | 0.424 | 0.417 | 4 |

As shown in Figure 2, during the period from 2001 to 2020, the TFEGE in China generally showed a steady decline. During the research period, the TFEGE dropped from 0.630 to 0.553, with an average decrease of 0.68% per year. The TFEGE of all regions in China also showed a downward trend, but the rate of decrease varied significantly across different regions. In the Eastern region, the decline in TFEGE was slight, and the trend was very stable. During the research period, the TFEGE fluctuated between 0.820 and 0.778, with an average annual decrease of 0.27%. The northeast region showed a strong downward trend, with noticeable fluctuations. The TFEGE dropped significantly from 0.791 to 0.307 during the research period, with an average annual decrease of 4.85%. The Western region displayed a downward trend in TFEGE, but the trend was relatively stable. Over the course of the research period, the TFEGE in the western region declined from 0.512 to 0.486, with an average annual decrease of 0.27%. The central region also showed a downward trend in TFEGE. During the research period, its TFEGE declined from 0.451 to 0.424, with an average annual decrease of 0.32%.

Tables 2 and 3, respectively, present the rankings of provincial-level and regional-level TFEGE. The ranking of each province and region is based on the average value of its TFEGE. In terms of province, Beijing, Guangdong, Hainan, Qinghai, and Ningxia were highly efficient throughout the entire research period. The national average TFEGE is 0.578. The average TFEGE values of Tianjin, Shanghai, Jiangsu, Zhejiang, and Fujian are all higher than the national average TFEGE. Throughout the entire research period, these provinces have maintained higher levels of TFEGE. Their input–output gaps are smaller, which gives them significant advantages in improving their TFEGE. In contrast, the average TFEGE of the other 20 provinces is below the national average TFEGE. During the entire research period, these provinces have lower TFEGE levels. Their inputs and outputs are not well matched, putting them at a disadvantage when it comes to improving their TFEGE. It can be observed that there are significant differences in TFEGE among the provinces in China. Provinces with high levels of TFEGE are primarily concentrated in the eastern and northeastern regions, whereas provinces with low levels of TFEGE are predominantly in the central and western regions.

From a regional perspective, the TFEGE of each region is unsatisfactory. The Eastern region exhibits higher TFEGE levels, with an average TFEGE significantly above the national average. However, other regions have lower TFEGE levels, with average values below the national TFEGE average. At the same time, the distribution pattern of regional TFEGE appears as follows: eastern region > western region > northeast region > central region. This distribution aligns with the implementation of economic reforms and the opening-up policy, indicating that the spatial distribution of TFEGE is not random. This inspires us to use the Moran Index to reveal the spatial pattern and relationships of TFEGE. Given that the Eastern region is the most developed area in China, the ranking of TFEGE suggests a certain correlation between TFEGE and a region's level of development. Developed regions possess ample funds, advanced technologies, rational industrial structures, and complete infrastructure. Furthermore, people in these regions tend to pay more attention to the quality of the ecological environment and social welfare compared to those in developing regions. These factors could all potentially be key to improving energy green efficiency. Therefore, to explore the factors influencing TFEGE, this study conducted further spatial econometric analysis.

## 5.2. The Spatial Autocorrelation of Total-Factor Energy Green Efficiency

Table 4 lists the Moran's *I* values for TFEGE. Throughout the study period, the values of Moran's *I* are all greater than zero and passed the significance test. Thus, we conclude that the TFEGE of the 30 provinces has a positive spatial autocorrelation, meaning that TFEGE exhibits a positive spillover effect. High (low) values are relatively concentrated among neighboring provincial units, displaying a strong spatial clustering pattern. The trend of Moran's *I* is unstable, with noticeable fluctuations throughout the research period. Between 2001–2004, the Moran's *I* showed an upward trend, peaking in 2004. From 2005–2008, the Moran's *I* index showed a downward trend. The Moran's *I* index rebounded in 2009, and it increased from 2009 to 2011. However, in 2015, the Moran's *I* index rapidly declined, reaching its lowest point. After 2015, it fluctuated up and down. This reveals that the spatial distribution of TFEGE across the 30 provinces exhibits instability and susceptibility to change.

**Table 4.** Total-factor energy green efficiency Moran's *I* test (2001–2020).

| Years | I | Z | P | Years | I | Z | P |
|-------|-----|-----|-----|-------|-----|-----|-----|
| 2001 | 0.317 | 2.787 | 0.005 | 2011 | 0.315 | 2.802 | 0.005 |
| 2002 | 0.320 | 2.814 | 0.005 | 2012 | 0.302 | 2.694 | 0.007 |
| 2003 | 0.332 | 2.911 | 0.004 | 2013 | 0.287 | 2.573 | 0.010 |
| 2004 | 0.344 | 3.018 | 0.003 | 2014 | 0.247 | 2.247 | 0.025 |
| 2005 | 0.288 | 2.589 | 0.010 | 2015 | 0.232 | 2.133 | 0.033 |
| 2006 | 0.272 | 2.466 | 0.014 | 2016 | 0.276 | 2.485 | 0.013 |
| 2007 | 0.276 | 2.497 | 0.013 | 2017 | 0.275 | 2.480 | 0.013 |
| 2008 | 0.264 | 2.406 | 0.016 | 2018 | 0.227 | 2.107 | 0.035 |
| 2009 | 0.289 | 2.596 | 0.009 | 2019 | 0.247 | 2.270 | 0.023 |
| 2010 | 0.307 | 2.738 | 0.006 | 2020 | 0.251 | 2.291 | 0.022 |
| The 10th Five-Year Plan (2001–2005) | | | | | 0.350 | 3.062 | 0.002 |
| The 11th Five-Year Plan (2006–2010) | | | | | 0.285 | 2.570 | 0.010 |
| The 12th Five-Year Plan (2011–2015) | | | | | 0.285 | 2.556 | 0.011 |
| The 13th Five-Year Plan (2015–2020) | | | | | 0.263 | 2.385 | 0.017 |
| Means (2001–2020) | | | | | 0.290 | 2.604 | 0.009 |

While the Moran's *I* index can reflect the spatial autocorrelation of TFEGE across 30 provinces, it has certain limitations. When some provinces exhibit positive spatial autocorrelation and others negative, the influences of these two types of provinces may counteract each other. In such cases, the Moran's *I* index might approach zero and display non-spatial autocorrelation. Therefore, to more accurately reflect the spatial characteristics and degree of agglomeration of TFEGE across 30 provinces, we have visualized it based on the Moran's *I* scatter plot using ArcGIS software version 10.8. We selected the Moran's *I* scatter plots of TFEGE for four periods (the periods of the 10th Five-Year Plan, 11th Five-Year Plan, 12th Five-Year Plan, and 13th Five-Year Plan), as shown in Figure 3.

According to the definition of the Moran's *I* scatter plot, it is divided into four quadrants. The first quadrant represents high–high clustering (H-H), where provinces with high TFEGE (above average) are surrounded by adjacent provinces with high TFEGE. The second quadrant represents low–high clustering (L-H), where provinces with low TFEGE are surrounded by adjacent provinces with high TFEGE. The third quadrant represents low–low clustering (L-L), where provinces with low TFEGE are surrounded by neighboring provinces with low TFEGE. The fourth quadrant represents high–low clustering (H-L), where provinces with high TFEGE are surrounded by provinces with low TFEGE. The first and third quadrants show positive spatial autocorrelation characteristics, while the second and fourth quadrants show negative spatial autocorrelation characteristics.

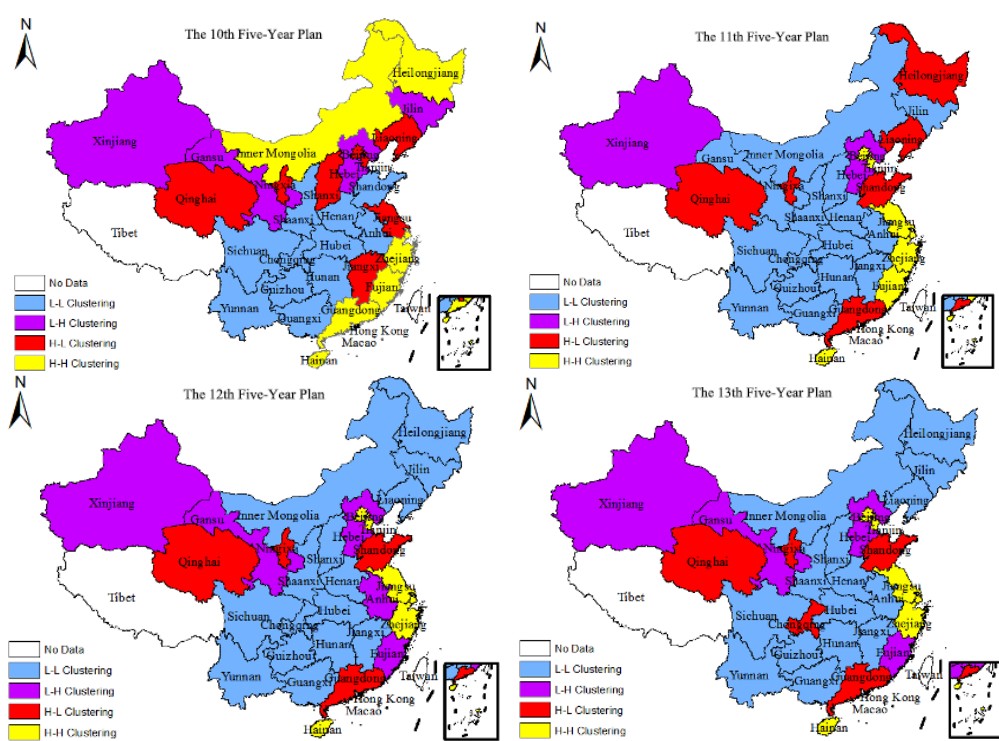

**Figure 3.** TFEGE Moran's *I* scatter plot.

Most provinces present H-H and L-L clustering, exhibiting positive spatial autocorrelation. For the four periods, 18 provinces (60%), 22 provinces (73.33%), 21 provinces (70%), and 21 provinces (70%) are located in the first and third quadrants, respectively. There are fewer provinces in the H-H quadrant, and they are mainly distributed in the eastern coastal regions. Over time, the provinces in the H-H quadrant gradually stabilize. The number of provinces in the L-L quadrant is greater, and is primarily concentrated in the central and western regions. Over time, the number of provinces in the L-L quadrant has increased, but the growth rate is not significant. The results further support the clustering feature. However, some provinces show H-L and L-H clustering and exhibit negative spatial autocorrelation.

For the four periods, 12 provinces (40%), 8 provinces (26.67%), 9 provinces (30%), and 9 provinces (30%) are located in the second and fourth quadrants, respectively. The number of provinces in the H-L quadrant is fewer and scattered across different regions, and over time, the number of provinces in the H-L quadrant has decreased. There are fewer provinces in the L-H quadrant, and they are scattered across different regions. Over time, the number of provinces in the L-H quadrant has gradually stabilized. This indicates that the spatial correlation of TFEGE in the 30 provinces not only exhibits spatial dependence, but also spatial heterogeneity.

### 5.3. Factors Influencing the Total-Factor Energy Green Efficiency

Based on the above analysis, we can see that there is significant spatial correlation and dependence in the TFEGE of the 30 provinces. However, using an ordinary regression model might lead us to underestimate or overestimate some factors. Therefore, this study chooses a spatial econometric model that can consider spatial effects to test the influencing factors of TFEGE in the 30 provinces. The choice of the spatial econometric model requires a Lagrange Multiplier (LM) test. Therefore, this study uses MATLAB software for the LM test. The results are shown in Table 5.

**Table 5.** OLS estimates and test results.

| Variable | Coefficient | t | Variable | Coefficient | T |
|----------|-------------|-----|----------|-------------|-----|
| ln TI | −0.233 *** | −13.018 | ln FDI | 0.036 *** | 2.816 |
| ln IS | 0.160 *** | 3.036 | ln EA | −0.060 *** | −4.444 |
| ln ES | −0.279 *** | −6.675 | ln GZ | −0.003 | −0.317 |
| ln TP | 0.079 *** | 5.326 | | | |
| LM-lag | | 5.399 ** | LM-error | | 0.072 |
| Robust LM-lag | | 11.303 *** | Robust LM-error | | 5.977 ** |

Notes: ***, ** indicate the 1%, 5% level of significance, respectively.

As shown in Table 5, the t-value for LM-lag is 5.399, which passes the significance-level test. But the t-value for LM-error is only 0.072, which does not pass the significance-level test. Therefore, based on the Anselin criterion, we choose the spatial lag model to test the influencing factors of TFEGE. According to the Anselin criterion, this study carries out tests for no space effects, spatial fixed effects, time fixed effects, time and space double fixed effects and random effects. Through comparative analysis, we screen out the most reasonable model. Table 6 shows the results of the five effects. The determination coefficient of the spatial lag model with spatio-temporal double fixed effects is 0.901, and the Log-L value is 482.154, both of which are higher than the others. At the same time, W*dep.var passes the 1% significance level test, showing significant spatial dependence. Finally, this study chooses the spatial lag model with time and space double fixed effect to test the influencing factors of TFEGE.

**Table 6.** SLM model estimation and inspection.

| Variable | No Space Effect | Spatial Fixed Effect | Time Fixed Effect | Time and Space Double Fixed Effect | Random Effect |
|----------|-----------------|----------------------|-------------------|------------------------------------|---------------|
| ln TI | −0.256 *** | 0.070 *** | −0.294 *** | 0.149 *** | 0.014 |
| ln IS | 0.199 *** | 0.016 | 0.202 *** | −0.137 * | 0.000 |
| ln ES | −0.276 *** | −0.126 *** | −0.237 *** | −0.193 *** | −0.111 *** |
| ln TP | 0.095 *** | 0.072 *** | 0.116 *** | 0.103 *** | 0.050 *** |
| ln FDI | 0.033 *** | −0.013 | 0.039 *** | −0.020 ** | −0.006 |
| ln EA | −0.070 *** | −0.166 *** | −0.092 *** | 0.043 | −0.101 *** |
| ln GZ | −0.007 | −0.017 ** | −0.016 | −0.026 ** | −0.021 *** |
| W*dep.var | −0.126 ** | 0.232 *** | −0.141 ** | 0.227 *** | 0.256 *** |
| R-squared | 0.564 | 0.891 | 0.584 | 0.901 | 0.881 |
| log-likelihood | 128.720 | 458.207 | 132.648 | 482.184 | −2013.521 |

Notes: ***, ** and * indicate the 1%, 5% and 10% level of significance, respectively.

(1) Transportation Infrastructure. The coefficient of transportation infrastructure is significantly positive and has a positive impact on TFEGE. With each 1% increase, the result of TFEGE will decrease by 0.149%. The construction of transportation infrastructure such as railways and highways can greatly promote interconnectivity between provinces. This is consistent with the conclusion of Song et al. [45]. Moreover, the development of transportation infrastructure can effectively promote technology exchange, commodity circulation, industry transfer, and reduction of policy learning costs, thereby directly enhancing the interactive intensity of TFEGE.

(2) Industrial Structure. The industrial structure coefficient is significantly negative and has a negative impact on TFEGE. With each 1% increase, the result of TFEGE will decrease by 0.137%. The industrial structure of most provinces in China is still dominated by the secondary industry, which includes many high-pollution, high-consumption brown industries. This is consistent with the conclusion of Lv, Yu and Bian [31]. Obviously, due to the large number of these industries, the improvement in TFEGE is impacted negatively.

(3) Energy Structure. The energy structure coefficient is significantly negative, and it has a negative impact on TFEGE. Coal, as a non-clean energy source, not only has low thermal efficiency but also produces a large amount of pollutant gases, causing serious environmental pollution problems. Therefore, an increase in the proportion of coal consumption is not conducive to the improvement of TFEGE.

(4) Technological Progress. The coefficient of technological progress is significantly positive and has a positive impact on TFEGE. The invention and creation of new technologies, the dissemination of new knowledge, and the technological innovation by R&D input have significantly improved regional technological capabilities, providing good technical support for regional scientific and technological progress and energy conservation and emission reduction.

(5) Foreign Direct Investment. The foreign direct investment (FDI) coefficient is significantly negative and has a negative impact on TFEGE. With each 1% increase, the result of TFEGE will decrease by 0.020%. FDI can no longer bring significant technology spillover effects to each province as it used to. In addition, Yan and Qi [46] found that FDI is one of the main factors that causes an increase in PM2.5 concentrations. At the same time, when each province accepts foreign investment, it needs to passively abide by the relevant regulations of foreign investment. These are the main reasons why FDI cannot promote TFEGE.

(6) Economic Agglomeration. The EA coefficient of economic agglomeration is positive, but not significant. In the process of economic agglomeration, accompanied by various positive externalities, economies of scale, cost savings, and technology and knowledge spillover effects will all contribute to the improvement of TFEGE. However, due to the insufficient concentration of the economy in China's provinces, there has not been a significant positive impact.

(7) Environmental Regulation. The environmental regulation coefficient is significantly negative, which has a negative impact on TFEGE. With each 1% increase, the result of TFEGE will decrease by 0.026%. Theoretically, the government can improve features such as resource wastage and environmental pollution caused by market blindness and irrationality and pursue maximum benefits through relevant means. However, according to the regression results, TFEGE has not improved with the strengthening of government regulation. The reason is that environmental governance investment is a "passive" behavior. Enterprises blindly pursue economic benefits, ignore resource waste, environmental pollution and other issues, and cannot implement government-led treatment measures and policies.

## 6. Conclusions and Implications

This study proposes the concept of total factor energy green efficiency (TFEGE) and constructs a total factor energy green efficiency evaluation index system covering economic, environmental and social dimensions. This indicator system incorporates relevant indicators of the social dimension, improving the comprehensiveness and comprehensiveness of the measurement results. In the future, it can provide a new evaluation system for regional energy sustainable development evaluation research. At the same time, energy green efficiency is the optimization of traditional energy efficiency. It can also provide a new idea for energy efficiency research in the future.

This study builds the UN-SLM-DEA model to measure TFEGE in 30 provinces in China. This is used to clarify the true level of TFEGE in 30 provinces in China. The UN-SLM-DEA model can improve the accuracy and scientificity of efficiency measurement results. It may help policymakers to clarify the development status and formulate targeted measures. It can also provide new methods for energy sustainable development evaluation in the future.

This study uses the global Moran's I index and Moran's I scatter plot to analyze the spatial characteristics of TFEGE. It explores the spatial distribution differences of TFEGE in 30 provinces in China. The space–time double fixed effects SLM model is

introduced to examine the influencing factors of TFEGE in order to scientifically identify the improvement-driving force of TFEGE. The spatial econometric model can effectively solve the shortcomings of traditional econometric models that ignore spatial effects and ensure that the driving force for improvement can be accurately identified. It can provide a new research tool for the study of influencing factors in the future.

Through the above research, this article draws the following conclusions.

(1) Based on the average TFEGE of 30 provinces in China from 2001 to 2020, only Beijing, Guangdong, Hainan, Qinghai, and Ningxia are efficient. Other provinces' TFEGE performances are not satisfactory, and there is still room for improvement. According to the average TFEGE in the four regions of China from 2001 to 2020, none have reached efficiency. The TFEGE in each region presents the following pattern: east > west > northeast > central. This distribution is consistent with the implementation of economic reforms and the opening-up policy.

(2) TFEGE of the 30 provinces shows a positive spatial autocorrelation, indicating a strong spatial clustering pattern. The spatial distribution of TFEGE in the 30 provinces is unstable and prone to change. Provinces in the H-H quadrant are mainly distributed in the eastern coastal regions, and provinces in the L-L quadrant are mainly situated in the central and western regions. The spatial correlation of TFEGE in the 30 provinces not only shows spatial dependence but also spatial heterogeneity.

(3) The influencing factors of TFEGE include transportation infrastructure, industrial structure, energy structure, technological progress, foreign direct investment, and environmental regulations. Transportation infrastructure and technological progress have a positive effect on TFEGE, while industrial structure, energy structure, and foreign direct investment can have a negative impact, thereby reducing TFEGE.

Based on the above conclusions, the following recommendations are proposed.

(1) Enhance research on the implications of TFEGE, encourage scholars to conduct studies considering social development, prosperity, and green development. Changing traditional perceptions of energy efficiency through the academic community's outreach to policymakers, businesses, and the general public can encourage consideration of sustainable energy.

(2) Strengthen infrastructure construction, control population growth, promote urbanization, prioritize science and education, cultivate high-quality talents, promote the development of medical and health services, enhance social comprehensive service capabilities, improve people's livelihoods, and achieve the goal of improving TFEGE.

(3) Provinces with higher levels of TFEGE should maintain a stable social status and consolidate achievements. Furthermore, they should enhance their industrial, technological, and social service levels to achieve a triple win in energy efficiency, ecological environment, and social welfare.

(4) Provinces with lower levels of TFEGE need to improve their infrastructure, develop the economy, and attract funds, technology and talents through economic prosperity. It is also essential to pay attention to comprehensive social development, ensuring high-quality policy implementation. Simultaneously, these provinces should fully absorb advanced experiences from developed regions and improve regional TFEGE through joint efforts in various aspects.

**Author Contributions:** Y.L. and C.L. contributed to the conceptualization, methodology, investigation and writing—original draft. Y.L. and C.L. performed research model, data collection, data curation and formal analysis. M.-J.L. participated in the manuscript revision, review, editing and validation. All authors have read and agreed to the published version of the manuscript.

**Funding:** This research received no external funding.

**Institutional Review Board Statement:** Not applicable.

**Informed Consent Statement:** Not applicable.

**Data Availability Statement:** Not applicable.

**Acknowledgments:** The authors would like to thank the editors and anonymous reviewers for their insightful comments and suggestions.

**Conflicts of Interest:** The authors declare no conflict of interest.

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
