# Peer review of "A Study on the Measurement and Influences of Energy Green Efficiency: Based on Panel Data from 30 Provinces in China"

_sustainability, doi:10.3390/su152115381_

Round 1

Reviewer 1 Report

Comments and Suggestions for Authors

Characteristics and contributions of the paper:

-          The contribution is actual due to the sustainable development and presently emphasized green economy, especially in such economy as in China, and therefore convenient for the publication in the journal.

-          The abstract is structured and makes it easy for readers to orientate in the paper

-          The contribution of the paper is at two levels – national and regional. Regional development is also very important for the sustainability

-          The results of the paper show which regions are ineffective from the view of analyzed indices, as well as determination of the individual indicators impact to the found situation. This bring variety of results use in the practice.

-          Literature review consists of the literature research of individual analyzed area, together with determination of the research gap in the literature. Authors used various subjects of the research, supported by the limitation of the research due to the data availability. Limitation of the research is underlined by selection the research methods. By this way paper presents scientific publication.

-          The research is gradually extended by consequent analyzed areas, connected with each other. The single research is made in number of China provinces, moreover the research is made in quite long period, which provide good comparison.

-          The results of the paper can be used in broad area, due to the number of analyzed periods – for time development, due to the number of regions – for regional sustainable development, etc. The results are individually separated in the paper, due to the publication in the journal Sustainability part 3.Energy Structure and part 7.Environment Regulation are most proper.

Recommendations for paper improvement:

-          I recommend to check English grammar

-          The title of the contribution is very long, I suggest to use abbreviation for well-known indicator

-          In the introduction authors described analyzed areas, here I would recommend to support their state by literature, for example – Csikosova, A., Janoskova, M., Culkova, K. Providing of tourism organizations sustainability through triple bottom line approach.  Entrepreneurship and Sustainability Issues, 8(2), 764-776. DOI: 10.9770/jesi.2020.8.2(46)

-          I recommend to use lower index for CO2 – page 2, the same page 8, table 1 – km2

-          The goal, determined at the end of the literature review is too abstract; I recommend clarifying the goal in the introduction, not only in the part Methods.

-          Part Methods consists of number of equation, I recommend using in the text to state indexes by italics font due to the readability increasing, also some indexes are too small to be readable, see for example page 6, last sentence in 3.2.2.

-          Numbering of equation must be aligned

-          The analyzed period is 2001-2020, however all illustrations present period 2002-2020 – (page 8 is also year 2000) - please, unify. The task is also in Table 4 – why five year plans for 2001-2005, etc., , when presently is 2023?

-          Figure 1 – not china, but China

-          Please, mention the source of the illustrations

-          Legend for Figure 1 should be enlarged due to the readability

-          Figure 2 – if the print version of the paper would not be in color, please use different type of curves

-          Title of figure 3 is not understandable – please, clarify

-          Not all abbreviation are known for the readers, for example OLS – table 5

-          In conclusion and implication I recommend to separate recommendations 1-4 to individual paragraphs

-          I recommend to check formatting of the references – sometimes journals are mentioned as abbreviations, sometimes as whole title of the journals

-          I recommend to use in references also DOI (if possible) and web side of database used

Comments on the Quality of English Language

I recommend to check english grammar by english editor.

Reviewer 2 Report

Comments and Suggestions for Authors

The study entitled “Investigation into the measurement and influences of total-factor energy green efficiency within the framework of spatial perspective: Based on panel data from 30 provinces in China” e deployed the SBM-DEA model, using energy, capital and labor as input indicators, and GDP and social dimension metrics as desirable outputs, and "three wastes" as undesirable outputs, to assess the TFEGE across 30 provinces in China from 2001 to 2020. Here are my comments to the authors:

1)     Please add the objective, novelty and significance of this research in the last paragraph of introduction.

2)     “The research content extends from the economic field to the environmental field and then further to the social field. These efficiencies align with social development requirements and are products of social progress” is repeated in the text.

3)     Please add some quantitative results to the abstract.

4)     The captions of figures should be written bellow the figures.

5)     Please add some explanation about SLM model estimation and inspection.

6)     The authors have not described the research backgrounds comprehensively and widely. Adding some new up to dated references about the energy strategy efficiency and spatial 3 perspective is suggested like:

a) “Potential assessment of renewable energy resources and their power plant capacities in Iran”

b) “Assessment of renewable energy production capacity of Asian countries: a review”

7)     To increase the reliability of the research papers, sentences in the paper should be written based on the objective contents using the previous study including the scientific basis rather than the author’s subjective views.

8)     Variable description should be written as “table 1” in the text.

9)     The overall results and discussions need improvement. I suggest the authors discuss what the results of the analysis imply and why the results have come up in a certain way. For instance, what are the future perspectives?

Comments on the Quality of English Language

I recommend authors proofread the entire manuscript once again to make sure there are no grammatical mistakes.

Round 2

Reviewer 2 Report

Comments and Suggestions for Authors

There are still a lot of issues unsolved in this paper. First, the novelty is still unclear. The authors mentioned a newly developed spatial statistical model is used to examine the influencing factors and spillover effects but the used method is not new and there are many previous papers which used this method. The captions of figures are still written above the figures. SLM model estimation and inspection and its use in this paper is vague. The literature review is very weak, suggested references and comment 6 of the previous review still remains. The paper is not qualified to be published in Sustainability in this form.

Comments on the Quality of English Language

The English is good.

Round 3

Reviewer 2 Report

Comments and Suggestions for Authors

The raised issues are resolved and the paper can be accepted in its current form.